# Exploring the content of the STAND-VR intervention: A qualitative interview study

**David Healy[1]\*, Emma Carr[2], Owen Conlan[3], Anne C. Browne[4], Jane C. Walsh[1]**

**1** School of Psychology, University of Galway, Ireland, **2** Amara Therapeutics, Galway, Ireland, **3** School of Computer Science and Statistics, Trinity College Dublin, Ireland, **4** School of Medicine, University of Galway, Ireland

\* d.healy24@nuigalway.ie

**Data Availability Statement:** Supplementary data has been deposited in OSF and can be found at https://osf.io/fuegq/?view_only= b475a5ceb5bd40519f7510ebe2a72940. All other

## Abstract

Prolonged sedentary behaviour has been identified as a potential independent contributor to a number of chronic conditions as well as mortality. The integration of digital technology into health behaviour change interventions has been shown to contribute to increases in physical activity levels, reductions in time spent sedentary, reductions in systolic blood pressure and improvements physical functioning. Recent evidence suggests that older adults could be motivated to adopt a technology such as immersive virtual reality (IVR) due to the added agency it can potentially afford them in their lives through physical and social activities offered in IVR. To date, little research has attempted to integrate health behaviour change content into an immersive virtual environment. This study aimed to qualitatively explore older adults' perspectives on the content of a novel intervention, STAND-VR, and how it could be integrated into an immersive virtual environment. This study was reported using the COREQ guidelines. Twelve participants aged between 60 and 91 years took part. Semi-structured interviews were conducted and analysed. Reflexive thematic analysis was the chosen method of analysis. Three themes were developed, "Immersive Virtual Reality: The Cover versus the Contents", "Ironing Out the (Behavioural) Details" and, "When Two Worlds Collide". These themes offer insights into how retired and non-working adults perceived IVR before and after use, how they would like to learn how to use IVR, the content and people they would like to interact with and finally, their beliefs about their sedentary activity and using IVR. These findings will contribute to future work which aims to design IVR experiences that are more accessible to retired and non-working adults, offering greater agency to take part in activities that reduce sedentary behaviour and improve associated health outcomes and, importantly, offer further opportunity to take part in activities they can ascribe greater meaning to.

## Author summary

Sedentary behaviour is defined as any waking behaviour that takes place in a sitting, lying, or reclining position each day while exerting little to no effort. Six or more hours of time spent sedentary each day has been associated with the development of a number of

data are in the manuscript and Supporting Information files.

**Funding:** This work was conducted with the financial support of the Science Foundation Ireland Centre for Research Training in Digitally-Enhanced Reality (d-real) under Grant No. 18/CRT/6224. The funders had no role in study design, data collection and analysis, decision to publish, or preparation of the manuscript.

**Competing interests:** The authors have declared that no competing interests exist.

chronic conditions as well as mortality. Older populations tend to spend prolonged periods of time sedentary each day. Immersive virtual reality (IVR), a relatively new digital technology, offers new ways to be less sedentary which retired and non-working adults can potentially ascribe more meaning to, such as taking part in physical activities they enjoy as well as facilitating social connection. IVR is a computer technology that makes a person feel like they are somewhere else. The findings from this study describe how retired and non-working adults perceived IVR before and after use, how they would like to learn how to use IVR, the content and people they would like to interact with and finally, their beliefs about their sedentary activity and using IVR. These findings will inform the design of future virtual experiences that are tailored to retired and non-working adults' needs and preferences.

## Introduction

Maintaining health and wellbeing into old age has become a priority in recent years as the number of people over the age of 65 rapidly increases [1]. Prolonged sedentary behaviour has been identified as a potential independent contributor to a number of chronic conditions as well as mortality [2–4]. Sedentary behaviour is defined as any waking behaviour which involves expending $\leq 1.5$ metabolic equivalents while in a sitting, lying or reclining position [5]. Although clinical guidelines are yet to be established for prolonged sedentary behaviour, epidemiological evidence suggests that 6 or more hours of sedentary behaviour per day is associated with numerous morbidities and all-cause mortality as well as significant costs to public healthcare services [6,7]. Prolonged sedentary behaviour can be defined in a number of different ways. For this study, we focused on what point a cumulative number of sedentary bouts, accrued throughout a single day, are correlated with negative health outcomes over time– which is approximately 6-hours [6]. A sedentary bout can be described as a period of uninterrupted time spent in a sedentary position (i.e., sitting, lying, or reclining while awake) [5]. A conservative estimate of a single objectively measured sedentary bout is 10-minutes, with most sedentary behaviours ". . . observed within bout durations of <10 minutes. . ." [8]. Carson and colleagues [9] specifically reported that $\geq$20-minute prolonged sedentary bouts could be particularly harmful to adults with their study reporting associations between bouts of this length and higher insulin and lower diastolic blood pressure levels, in a large sample of 4935 adults aged between 20–79 years. Other findings from this study suggested that each additional 10 breaks per day were associated with health outcomes such as higher HDL-cholesterol and lower insulin levels, among other outcomes. They concluded that breaking up these bouts throughout the day could mitigate negative health outcomes such as high insulin and low diastolic blood pressure.

Systematic review evidence indicates that many older adults are spending greater than 6-hours per day in a sedentary position [7,10]–with objectively measured sedentary activity revealing that older adults are spending an average of 9.4 hours sedentary each day [10]–bringing them over the threshold for potential health risks that could impede their overall health and wellbeing into old age.

In the past 30 years, digital technology has become a central part of our lives, changing the way we approach many of today's problems, including health promotion and behaviour change. Novel approaches to designing digital behaviour change interventions have been established to support the integration of health behaviour change content with digital technologies [11]. The person-based approach is one such approach which places the end-user at the

centre of the design process and recommends iteratively designing suitable health behaviour change content that can be integrated with digital technology based on continuous participant feedback. Instances of intervention development using the person-based approach offer insight into the potential of digital technology in user-centred health behaviour change intervention development [12,13]. Past examples of studies which integrate digital technology with health behaviour change interventions have shown to increase physical activity levels, reduce time spent sedentary, reduce systolic blood pressure, and improved physical functioning [14].

Immersive virtual reality (IVR) can be defined as fully computer-generated environments that are displayed through a head-mounted display [15]. A synthesis of recent qualitative studies exploring older adults' experiences and perceptions of IVR indicated that older adults could be motivated to adopt a technology such as IVR due to the added agency it can potentially afford them in their day-to-day lives [16]. This materialises in the form of various physical activities that are available in IVR, opportunities to travel to places around the world where it may otherwise be impossible to do so, connect with others who may not be available to meet physically through the embodiment of virtual avatars [17], as well as a variety of other meaningful experiences [16].

To our knowledge, little research has attempted to integrate health behaviour change content into an immersive virtual environment. Based on evidence now suggesting that older adults would be motivated to use such a technology [16], it is worth exploring if IVR could offer a new platform for digital health behaviour change interventions.

Using the person-based approach [11], an IVR behaviour change intervention was being developed by the study team while this study was being conducted. The behaviour change wheel guide to intervention development was utilised to develop the intervention content [18]. This process involved collating existing evidence from the literature to first understand the context within which prolonged sedentary behaviour takes place in older populations. After this understanding was established, a target behaviour was chosen to change; the determinants for which were identified and organised using the theoretical domains framework (TDF) [19]. The target behaviour chosen through this process was, taking part in meaningful non-sedentary activities in IVR. Through further identification of intervention functions and behaviour change techniques, intervention content was created [18]. This intervention content offers additional opportunities for retired and non-working adults to reduce the prolonged periods of time they spend sedentary each day using IVR. During the initial development of this content, it is important to understand the perspectives of the potential users with regards to the proposed content prior to integrating it with a virtual environment [11]. This study therefore aimed to explore retired and non-working adults' experiences with IVR, their views on the STAND-VR (SedenTAry behaviour iNtervention Development using Virtual Reality) intervention content, and their views on using IVR to help reduce their time spent sedentary.

## Methods

This study was reported using the Consolidated Criteria for Reporting Qualitative Research (COREQ) guidelines [20].

### Research team and reflexivity

**Personal characteristics.** The research team consisted of three health psychology researchers, one computer scientist and one general practitioner. The lead author (DH) conducted the interviews. At the time the interviews were conducted, DH had completed an undergraduate degree in applied psychology and a master's degree in health psychology. DH was a PhD student completing research in the field of health psychology. He was a 25-year-old

man. To date, DH had completed a systematic review and thematic synthesis exploring older adults' experiences and perceptions of IVR [16]. DH was a frequent user of ubiquitous digital technologies such as smart phones, personal computers, and activity watches. DH also had three years of experience using IVR technologies by the commencement of this study.

**Relationship with participants.**   No relationship was established with the participants prior to study commencement. Participants knew that the interviewer was a PhD student exploring if IVR could be used to support retired and non-working adults over the age of 55 to reduce their time spent sedentary.

**Patient and public involvement.**   A patient and public involvement (PPI) panel was formed to contribute to the design of the study. The panel consisted of two retired adults over the age of 55. During the study design phase, the PPI contributors were invited to consult on the development of the interview schedule (see S1 Text). Their feedback led to significant changes to the wording of the interview schedule, to make the questions clearer and more accessible to the general public. The PPI contributors were also consulted about the wording and appearance of the study advertisement flyer.

## Study design

**Theoretical framework.**   Reflexive thematic analysis was the chosen method of analysis for this study, providing an epistemologically and ontologically flexible approach to qualitative analysis. As a result, it is a method that can be used across a range of research contexts. The current study aimed to explore the content for a behaviour change intervention with retired and non-working adults. Therefore, this study is grounded in a critical realist ontology and contextualist epistemology [21], to allow for subjective meaning to be explored with each participant–such as their views on how comfortable the IVR equipment is or their preferences regarding goal setting–but also rooting this subjectivity within the context of a single reality [22]. This requires the researcher to interpret, to an extent, why each participant holds certain beliefs by considering forces such as cultural norms or physical capabilities. Furthermore, reflexive thematic analysis enables patterns to be generated across the data.

**Ethical statement.**   Ethical approval was granted for this study by the University of Galway Research Ethics Committee (application reference number: 2021.05.008). At the beginning of each interview session, participants were invited to read the participant information sheet which explained why this research was being conducted and what was involved in taking part. Once participants verbally confirmed that they understood everything on the participant information sheet and agreed to continue with the study, formal written consent was obtained from each participant.

**Participant selection.**   Purposive and convenience sampling as well as snowballing were employed to recruit participants for this study. The recruitment strategy and inclusion criteria were developed based on the aims of the study as well as the broader PhD research project. These are reported in Tables 1 and 2 below.

Potential participants were approached through several channels. A flyer was created containing information about the study, the lead author's contact information and a prompt to

**Table 1. Recruitment Strategy.**

| Stratification Category | Description |
| --- | --- |
| **Age** | 55–60, 60–65, 65–70, 75–85, 85–90, 90+ years of age. |
| **Sex** | An even split of men and women if possible. |
| **SES** | Where feasible and appropriate, recruit participants of varying SES. |

**Table 2. Inclusion Criteria.**

| Inclusion criteria | Description |
|---|---|
| **Sedentary behaviour levels** | Retired and non-working adults who self-report 6+ hours of sedentary behaviour per day who are physically capable of being non-sedentary. |
| **Mobility status** | Must be able to use the technology independently once in use. It is essential to have sufficient mobility in their hands to use the controllers. |
| **Work status** | Retired or non-working. |

make contact if interested. Members of the PPI panel assisted with recruitment by providing retirement organisations with the study flyer. Study flyers were posted on social media and national newsletters of various retirement organisations and other support groups for older adults. Twelve participants were included in the study. No participants who were contacted directly after providing their contact details refused to participate, and no participants dropped out. Recruitment continued until it was determined that information power had been reached [23].

**Setting.** Data collection took place in a spacious, ventilated room on the University of Galway campus. A second researcher was present during the first IVR activity to mitigate any risks of falling during this activity. It was decided after this interview, however, that a second researcher would not be necessary as there were no safety risks that required their assistance.

**Data collection.** The interview schedule was informed by systematic review data exploring older adults' experiences and perceptions of IVR [16], the TDF [19], and the wider literature relevant to the study. Participants were asked to complete the "sitting time" section of the International Physical Activity Questionnaire [24] (see S1 Table), to confirm they spend six or more hours each day sedentary, as well as a demographic questionnaire (Tables 3 and 4, and S2 Text). Four participants described themselves as "not working" and the reasons given included medical conditions, child rearing and caregiving. No repeat interviews were carried out. An audio recording device was used to collect the interview data, which was then transcribed verbatim. Interviews lasted between 33 and 86 minutes. Field notes were collected after each interview in the form of a reflexive journal.

Using IVR during the interview session required participants to wear the equipment and interact with the computer-generated environments displayed to them through the head-mounted display using the hand-held controllers. The Meta Quest 2 was used to facilitate participants' virtual reality experience [25]. A recent systematic review [16] indicated that Meta

**Table 3. Participant Demographic Information (1).**

| Participants | Age | Sex | Ethnicity | Retirement status | Time retired (years) |
|---|---|---|---|---|---|
| **PT001** | Early 60s | Female | White Irish | Retired | 3 |
| **PT002** | Early 60s | Female | White Irish | Not working | 24 |
| **PT003** | Early 60s | Male | White Irish | Retired | 4 |
| **PT004** | Early 70s | Male | White Irish | Retired | 7 |
| **PT005** | Late 60s | Female | White Irish | Retired | 6 |
| **PT006** | Late 70s | Female | White Irish | Retired | 12 |
| **PT007** | Early 70s | Female | White Irish | Retired | 5 |
| **PT008** | Mid 70s | Male | White Irish | Retired | 8 |
| **PT009** | Early 90s | Female | White Irish | Not working | 67 |
| **PT010** | Early 70s | Female | White Irish | Retired | 0.5 |
| **PT011** | Mid 70s | Female | White Irish | Not working | n/a |
| **PT012** | Early 70s | Female | White Irish | Semi-retired | 2-days per week |

**Table 4. Participant Demographic Information (2).**

| Participants | Place of residence | Education level | Living status | Total time spent sedentary (hours) |
|---|---|---|---|---|
| PT001 | Suburban | n/a | Alone | 9 |
| PT002 | Suburban | Nursing and Midwifery | With a pet | 8 |
| PT003 | Suburban | Masters | With a partner | 12 |
| PT004 | Suburban | Bachelors | With a partner | 10.5 |
| PT005 | Suburban | Diploma | Alone | 12 |
| PT006 | Urban | Secondary | With a family member | 10 |
| PT007 | Suburban | Nursing and Midwifery | With a partner | 9 |
| PT008 | Suburban | Masters | With a partner | 12 |
| PT009 | Urban | Secondary/ALMC in music | Alone | 10 |
| PT010 | Rural area | Secondary | With a partner | 13.5 |
| PT011 | Rural area | Secondary | Alone | 9 |
| PT012 | Rural village | Bachelors | With a family member | 12.5 |

Quest equipment was suitable to be used with older adult cohorts. Participants experienced a virtual environment co-developed by the lead researcher in collaboration with a human-computer interaction researcher and a games developer. The training environment was named VR FOUNDations (Virtual Reality Familiarisation envirOnment for older adUlts with aND without dementia). Images of this training environment can be found in Fig 1 below.

## Data analysis

One researcher (DH) coded and analysed the data in NVivo 20 [26], and circulated the results with the rest of the research team to be discussed and refined where necessary. In line with Braun and Clarkes' steps for reflexive thematic analysis, initial codes were first developed, after which similar codes were organised into clusters or groups and finally, these groups were organised to form candidate themes [21,27]. Where relevant, subthemes were created under these themes.

Themes were generated from participant data [28]. However, as the interview schedule was, in part, derived from the TDF, the knowledge generated was generally within the scope of the constructs that make up this framework. Member checking of the interpretations made by the lead researcher during the write-up of the analysis was not carried out as it contradicts the ontological and epistemological positioning of reflexive thematic analysis [28], in which the researcher's interpretations are made as a result of their subjective engagement with the collected data.

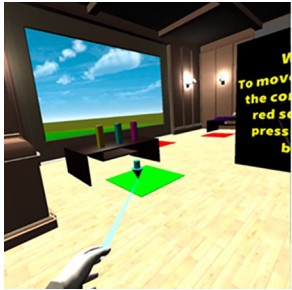 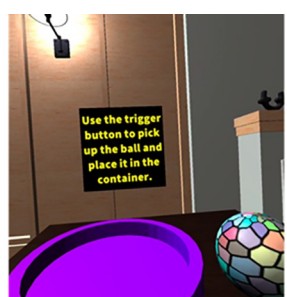 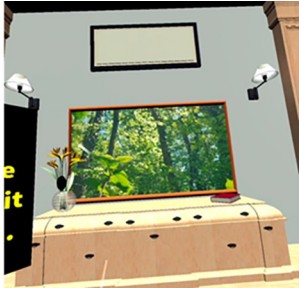

**Fig 1. Images of the VR FOUNDations Training Environment.**

## Results

### Summary of results

Three themes were developed, "Immersive Virtual Reality: The Cover versus the Contents", "Ironing Out the (Behavioural) Details" and, "When Two Worlds Collide". These themes explore retired and non-working adults' experiences with IVR, their views on the STAND-VR intervention content, and their views on using IVR to help reduce their time spent sedentary. As a large quantity of data was collected, the themes represent higher-level interpretations made during the analysis while design-specific findings are presented in the supporting information section (see S2 Table and S3 Table). These tables consist of design considerations for the STAND-VR virtual environment (S2 Table)–organised based on Abeele and colleagues' [29] design guidelines for IVR development for older adults and the TDF (S3 Table) [19].

### Reflexive thematic analysis

**Immersive virtual reality: The cover versus the contents.**   This theme illustrates participants' views on IVR prior to use followed by their views after experiencing it during the interview session. The majority of participants had never used IVR before, but the few who had recounted a positive experience with the technology:

"I find that the facility [a virtual environment] to be able to walk through your house, room by room, and, and the outside gardens, and upstairs and downstairs, an unbelievable experience to have".

A pattern identified while exploring participants' thoughts on IVR prior to use was their uncertainty around how this technology could be used in their own lives, particularly to reduce their sedentary behaviour:

"I think when I use it [IVR] maybe. . . I really have to use it to see what is [sic] going to give me or what benefit it will be for me, you know?".

Variations of this uncertainty were made by all participants prior to experiencing IVR. Some concerns were also raised prior to use, with a number of participants worried they would feel claustrophobic during the experience or would not be able to use the equipment. Despite these hesitancies, however, almost all participants were interested in finding out what IVR could offer as a tool to manage health, such as reducing sedentary behaviour, as well as a way of trying something that could be enjoyable. For example, prior to experiencing IVR during the session, one 72-year-old man with previous experience using IVR shared his thoughts on the potential for the technology, "it's unbelievably good. . . opens a mind-boggling sphere of opportunities for older people".

Participants' impressions after experiencing IVR were generally positive, with words such as "spectacular" and "fantastic" used to describe it. Participants saw IVR as a novel type of learning environment which brought with it the opportunity to broaden their imagination in new and challenging ways. Participants were pleasantly surprised by the capabilities of IVR, not realising it could offer such an immersive experience, "Because it could be so vivid. . . I wasn't expecting it to be. . . in as much detail as it was", with many even finding it difficult to put the experience into words. In contrast, participants also felt a technology such as this could be misused:

"I think it has the potential of, of wonderful stuff. It's no more than the internet. It could. . . bring [you] to the end of the world, it could also bring you torture".

The comparison to the internet here suggests that this participant is aware of IVR's potential for good but also cautious about what unforeseen adverse effects a technology like this could have.

After experiencing IVR, participants shared that they were intrigued by feelings such as presence, immersion, and the added agency which IVR could simulate. Almost all participants felt physically present in the virtual environment. This feeling brought about a range of different reflections, with participants awe struck by the experience, amazed that a digital technology could make them feel like they are somewhere else, "that you could feel you're, in a place that's 1000s of miles away, maybe? It really is. . . It's hard to believe that it can be done". The feeling of being physically present in the virtual environment was universally received as a novel experience that brought about a sense of wonderment and for some participants it was seen as an opportunity to escape reality for a while, like a form of respite or retreat.

Many participants also acknowledged the added agency IVR brought about by its interactive nature. The hand-held controllers gave participants a form of "power" over their actions in the virtual environment, enabling them to interact with objects in similar ways to the real world, such as picking up different objects and moving them to different places. A common pattern across the responses of participants who discussed this phenomenon was that the more freedom, or fewer limits, they were afforded to explore the virtual environment on their terms, the more enjoyable and meaningful the experience was. In contrast, some participants experienced a reduced sense of agency as a result of the novelty of the immersive experience–with participants nervous about falling as they did not trust their footing when immersed in the virtual environment and blinded from the real world.

Participants also shared why they would return to IVR in the future and the conditions under which they would do so. At the beginning of the interview session, it was made clear to each participant that the aim of this project was to explore if they would be interested in using IVR to reduce their time spent sedentary. However, after trying the technology, some participants were only interested in using IVR to take part in activities they could ascribe meaning to, rather than to reduce their time spent sedentary, "I would use virtual reality simply because I would enjoy [it]. It would be nothing to do with being sedentary". Participants also believed that persistent use would make them more proficient users of IVR, indicating that many believe this technology is one they can master. There were also a range of activities that participants were interested in experiencing in IVR (see S2 Table). Interestingly, a number of participants wanted these activities to be risky and stimulating, and more specifically, activities that they would be too frightened to do in reality. This feedback illustrates that participants see IVR as a means to broaden the range of activities they can take part in.

**Ironing out the (Behavioural) details.** This theme focuses primarily on how best the IVR experience can be facilitated for retired and non-working adults. Feedback from this part of the analysis was mapped onto domains of the TDF where appropriate (see S3 Table) and also organised into new interpretations that stand outside of this framework.

*Learning the Ropes.* This subtheme focuses on the preferred conditions under which retired and non-working adults would like to learn how to use IVR. Participants generally found the IVR learning curve easy, ". . .I found it okay. Nothing complicated about it, really". Most only needed a few minutes to become comfortable with the controllers and navigating the virtual environment. Most participants also indicated that they would prefer to learn how to use IVR with someone present who had experience using the technology. Participants generally felt this was necessary as a kind of "stand-by" support to make sure nothing goes wrong, rather than the need for any major assistance, ". . .you're kind of vulnerable and you want I suppose you want someone there just in case something happens". Participants gave up their sense of sight while in the virtual environment, making it critical for many of them to have someone physically present initially to get used to that experience. There was also wide support for accessible written instructions which users could reference if they were ever uncertain about any of the

features of IVR. The various combinations of instructions suggested by participants can be found in the supported information section (see S4 Table).

A number of participants were also interested in learning how to use IVR in a group environment. Group learning was seen as a motivator for them to learn how to use IVR as well as a more enjoyable means of doing so:

"I think maybe even presenting it to a group of people in a group. Because that way, you have a bit of interaction, and you can make it a fun thing".

Importantly, participants believed other people who were also learning how to use IVR could be more approachable than a more experienced, and supposedly younger, facilitator alone–indicating a need for an approach where experienced and less experienced IVR users learn together. This form of social support appeared to be important for some participants when considering how best to learn how to use IVR. Some participants noted that they would like a graded learning experience, with various conditions attached to these (see Table 2). Of note, the learning environment they were presented during the interview (VR FOUNDations) was deemed sufficient, suggesting a similar template could be used in future iterations to introduce participants to IVR. Some participants stressed that IVR is more appropriate for practical activities, referring to refining the skills required to interact with the virtual environment, rather than information gathering–referring to the passive consumption of information; whether it be in video, audio or text format in the virtual environment. This conflicted with some aspects of the designed intervention, which had proposed providing health information on sedentary behaviour in IVR. Tailoring the way information is delivered to participants is therefore an important element to highlight here.

*To Strategize or Not to Strategize.* This subtheme explores feedback on different types of strategies retired and non-working adults could use to take part in meaningful non-sedentary activities. It considers whether strategizing is something retired and non-working adults would be interested in doing and if so, what kind of strategies they would be interested in adopting.

When asked about their thoughts on using strategies to change their sedentary behaviour, there was a variety of responses given (see S3 Table). This illustrates how complex the topic of strategies is, with a clear need for them to be tailored to the individual. A common pattern across these suggestions was that they did not involve the use of digital technology–although some were open to this mode of delivery (see S3 Table).

Views on goal setting were divided (see S3 Table). Participants who were in favour of goal setting believed that having goals to take part in IVR activities would motivate them and would provide a sense of achievement when goals were successfully completed. In contrast, other participants believed that goal setting would take away from the IVR experience as it would make it competitive when they are not competitive people. Participants also believed that simply being able to use the technology and take part in activities they enjoyed was achievement enough, without any need to create a specific goal to do so.

Some participants were in favour of monitoring their sedentary behaviour, using techniques such as reminders to take part in IVR activities to break up their sedentary bouts. The ways in which participants imagined how these would materialise varied, however. Participants were open to both digital solutions such as on-screen reminders and a digital activity logbook that appeared in the IVR device (see S3 Table) as well as more traditional reminders such as internal self-monitoring or a physical timetable.

What was universal across the diverse range of strategies suggested, was that activities participants take part in would need to be done on their terms, referring to when, or if, they wanted to do it, rather than being told to take part by some form of external que, "No, it takes away the freedom to either use it or just get on with something else and then go to it at a time when you're completely relaxed. . .".

*Together is (mostly) Better.* This subtheme explores retired and non-working adults' thoughts on taking part in IVR activities with others. Furthermore, it explores how participants thought about being represented in the virtual environment and how they would interact with others in that environment.

A number of participants were not consciously aware of how they were represented in the virtual environment. Although they only embodied hands in the virtual environment, the common response to this was that they simply accepted this for what it was; it did not impede on their ability to interact with the virtual environment, "Oh, no, no. Once, once I got used to the idea [of just having hands], I actually forgot completely". Many participants added that the hands enabled them to interact with the environment and that was all that mattered to them–they were a "tool" that gave them more "power". Closely linked to agency discussed above, participants attributed their ability to successfully interact with the virtual environment, at least in part, to the representation of their hands. It represented the same way they would interact with physical objects in the real world and therefore appeared to be given value in the virtual world.

Although participants did not have the opportunity to engage with their own avatar or other avatars in a meaningful way in VR FOUNDations, when asked what they thought about the inclusion of avatars in the virtual environment in various ways, many were open to the idea (see S2 Table), stating they would be motivated to meet other people in IVR to share the experience together as it would be a way of enhancing social connection. In contrast, others believed an avatar was not necessary for individual experiences and even where it could be social, they thought it would probably lack the social cues to have meaningful social interactions with others embodied as avatars.

A number of participants suggested integrating a technology like IVR into community organisations. Some participants saw community organisations as a pivotal way to promote, teach how to use, and utilise IVR as another way of interacting with each other outside of meeting in person:

"...one sensible approach then would be to come to a group like ours [retirement organisation], where you have maybe 100 old people in one group... And you're getting a larger group of people who'd [at] least try it".

Community organisations were seen as an engine through which IVR could be introduced and adopted by retired and non-working adults in the community. They are seen as a safe environment for retired and non-working adults to learn how to use IVR with each other and eventually interact with each other in it.

*Offering and Influencing Opportunities to IVR Use.* This subtheme explores what opportunities IVR offer retired and non-working adults with regards to reducing sedentary behaviour–as well as many other opportunities. It also explores what would influence retired and non-working adults' ability to use the technology in the future. This subtheme bares strong links with the "opportunity" element of the COM-B model [18], exploring both social and physical barriers and enablers to IVR use.

The opportunities offered by IVR that were suggested by participants were diverse (see S2 Table). Participants generally saw it as a new tool to bring new positive change into their lives, offering them an escape from negative thinking, the opportunity to take part in activities they may be insecure to do in public, the opportunity to alleviate boredom and, an alternative to activities such as TV watching. The key in this theme is that participants saw opportunity with this technology. They saw the potential it had to enhance their lives in various ways that they would otherwise not have the opportunity to do:

"...in recent years, I have developed arthritis in my hands, which has affected my grip... So I can see if... I could play my tennis on that and play with other people and satisfy that competitive streak and then also be active".

With regards to opportunities influencing IVR use, all participants said they would have space in their home to set up the IVR equipment. The current price of the equipment was also not seen as an issue for those who commented on it. In general, when discussing any potential physical barriers to use with regards to opportunity, participants did not see any major ones. Some did stipulate, however, that they would require some social support when setting up the equipment. This is linked closely to previous comments made about needing someone else there in person to learn with–such as within a community organisation.

**When two worlds collide.** This theme explores retired and non-working adults' beliefs about sedentary behaviour and whether digital technology can or should be adopted by them to assist in reducing this behaviour.

*Constructing Beliefs and Identity*. This subtheme offers insight into how retired and non-working adults' sense of identity relates to their beliefs about health and digital technology. It offers insight into these beliefs and how their identity has been shaped by various events throughout their lives.

It was clear that almost all participants understood that sedentary behaviour has negative effects on health, with some indicating that it is a common-sense belief. They observed the negative impact of prolonged sedentarism in their own lives with some sharing that they feel unwell or depressed when they spend too much time sedentary. Participants were also affected by the negative health outcomes for others in their lives who lived sedentary lifestyles. For some participants, there was a belief that physical activity made up for their time spent sedentary–regardless of whether they were sedentary for prolonged periods of time each day:

"most of the group you're going to meet [participants from the same retirement organisation] are not all that [sedentary] they'll often be out walking quite a bit, they keep active".

In this case, the participant was aware of the benefits of physical activity but were unaware of the potential negative effects of prolonged sedentary behaviour, which each of the participants reported taking part in. These beliefs highlight the importance of conveying the difference between physical activity or inactivity versus prolonged sedentary behaviour or non-sedentary behaviour. Thoughts about future health were also highlighted as a key motivator to take part in non-sedentary activities in the present:

"I do kind of think, Gosh, in another 10 years what will I be like, will I be able to move at the same speed that I'm able to do at the minute?".

The formation of participants' identities in the context of their sedentary behaviour was also discussed during each interview. A pattern that developed across a number of participant accounts was the way time, and the age-related changes associated with time, shaped the way they identified with being sedentary or non-sedentary throughout their lives. Participants spoke about how consistent hard work across time instils and maintains non-sedentarism as part of their identity. They also shared how social support can help maintain such an identity and terms such as "retirement" attributed to them when they reach a certain age can be potentially stigmatising and influence their identity–with the suggestion that now is the time for them to "slow down" and rest more rather than being active. Additionally, closely linked to participants' beliefs about their ability to use IVR equipment was how this in turn formed part of their identity. The responses from participants about their beliefs about capabilities and identity in this case suggests a temporal arrangement of the two–with the belief first forming through the encouragement of their use of IVR, as well as their actual use of it, and later the formation of this instilled belief as part of their identity.

Participants expressed that simply trying IVR strengthened their beliefs about their capabilities in using it. For many, using IVR broke down the prior assumption that they would not have the ability to do so:

"I think, to reassure them [retired and non-working adults], that they they're capable of doing it, like people are afraid, oh that's too high tech for me. . . I think if they give it a go, and, you know, try it, and be open to change and open to new ideas".

An important stipulation made by some participants was that IVR needs to first be presented to them in an accessible way. Participants pointed out that it is prior fears and uncertainties about their capabilities that would influence their decision to try it, so presenting IVR in a positive and accessible way was considered important.

*Technology and Health*: *A Strange New World*. This subtheme explores participants' views and experiences with digital technology to date and what their preferences would be for delivering health information using digital technology, if at all.

Many participants shared their thoughts on their prior experience with digital technology, primarily discussing it in the context of health management. Some participants said they do not use digital technology to help manage their health as they would not have any meaningful use for it, and because they believed they are too old to benefit from it, "What's the point at this stage of my life?". Participants generally used digital technology to browse the internet and connect with others on social media. In contrast, others have found utility for digital technology to help manage their health, seeing it as means to give them more control and confidence. External factors influencing a few participants' views on digital technology use included having younger people, who were perceived as more tech-savvy, available to assist them in using it as well as the pandemic which began in 2019 and forced some to adopt digital technology to do things like staying in touch with others, "...my wife can Zoom [online communication medium] now where she could barely click on a mouse before this [pandemic]".

After experiencing IVR during the interview session, almost all participants indicated that they would be interested in trying it in the future to reduce their time spent sedentary. Participants' reflections on this point generally indicated that IVR would act as a motivator to take part in non-sedentary activities due to the experiences it would offer them as well as simply acting as an additional outlet to be non-sedentary. However, some participants showed preference to other forms of non-sedentary activity and remained unconvinced of the benefits IVR for their health, "Well, at the moment, I'm not convinced... of the health benefits of it, let's say". Other participants shared that they would need more experience with IVR before deciding on whether it is something they would use.

Delivery preferences for receiving information about sedentary behaviour were also discussed. This is linked to the TDF element, knowledge, which is a posited determinant of health behaviour change that emphasizes the importance of individuals first needing to know why and how they need to change their behaviour before going about changing it. The majority of participants were open to receiving health information in IVR in a variety of different forms (see S4 Table). Some participants liked the idea of another avatar presenting health information as it would be more interesting. In contrast, others preferred the idea of receiving health information via a video in IVR rather than from an avatar. Participants also suggested other preferences for health information delivery outside of IVR (see S4 Table). Beyond the mode of delivery itself, participants emphasized that it is important for the information to be accessible, transparent, reliable and actionable. Participants saw their doctor, family members and friends as people who they would trust to receive this information from. In general, the central concept formed across each of these suggestions was that a tailored approach to health information delivery inside and outside IVR is necessary to facilitate everyone's preferences.

## Discussion

### Main findings

During this study, retired and non-working adults were introduced to IVR and interviewed about this experience and their views on the STAND-VR intervention content. Through reflexive thematic analysis, three themes were generated which offered insights into how

retired and non-working adults perceived IVR before and after use, how they would like to learn how to use IVR, the content and people they would like to interact with and finally, their beliefs about their sedentary activity and using IVR.

## In the context of existing research

IVR was an enigma to participants who had never tried it before; they needed to experience it before they could grasp what it had to offer. All participants were enthusiastic, in some way, about their experience with IVR and found it to be an accessible technology to use once tried–providing novel immersive experiences many had never encountered before. The uncertainty experienced prior to use is evident in previous research, where older adults were quite negative in their preconceptions of IVR [30–32], while others, similar to sentiments in this study, were unsure what it would be like but were interested to try it nonetheless [33]. The evidence suggests a need to ensure that IVR is presented in a transparent and accessible way to older populations, where a lack of understanding about the technology is exhibited. In the current study, participants found IVR easy to use. This is also evident in previous research, where participants navigated the IVR equipment and virtual environments with ease and competence, which increased with practice [34]. This shift from uncertainty about IVR and personal capability, to mastery with little effort, was also reported in a systematic review exploring older adults' experiences and perceptions of IVR [16]. This change in perspective before and after use points to a temporal pattern evident in older adults' experiences with IVR, with a general sense of uncertainty around what IVR is and in some cases, a lack of self-confidence in their ability to use it. Current evidence also highlights that once older adults have had the opportunity to use it, IVR is generally received as an accessible technology that offers novel and meaningful immersive experiences.

An interesting pattern evident across the data was specific participant motives for using IVR in the future. Participants were more interested in the meaning they could ascribe to IVR beyond the health benefits of reducing their sedentary behaviour. Participants' primary interest in IVR was using it as a means of enjoyment rather than a means to reduce their time spent sedentary. Previous research has identified a similar phenomenon described as "incidental physical activity" [35]. This highlights that although the health-promoting activity may be conveyed formally as a health behaviour in the eyes of the intervention developer, for the participant, it is not seen this way; rather, the activity is simply something they enjoy, and any health-related outcomes are secondary for them.

A range of different instruction formats were recommended by participants regarding how they would like to learn about using IVR. The range suggested the importance of tailoring information provisions to the individual user's needs, to ensure the technology and the experience are accessible to all. A number of participants highlighted that they would like someone present to help them to set up the equipment. This suggestion is also reported in previous research, where some participants shared that they would like someone present to reassure them that they are safe [36]. The IVR experience means users are completely blinded from reality. A common pattern across the literature is that this feature can be disconcerting for older adults and requires them to have someone present while acclimatising to this new experience.

A notable finding was the belief that passively receiving information through IVR would not be an efficient use of the technology, with one participant claiming that other modes of delivery such as YouTube and other web browsers are more effective ways of acquiring such information. This is an important finding as part of the STAND-VR intervention will be exploring how participants find receiving health information in IVR–which may relate to

previous discussions about participants' preconceptions of IVR. Specifically, although participants did not have an opportunity to receive health information in VR FOUNDations, the general views were that it would not be a useful way to receive it. There is little evidence on how older adults receive health information in IVR, with most of the research focusing on other interactive and passive activities such as sports, travel, and reminiscence, to name a few [30,37,38]. Some participants were interested in the idea of receiving health information from an avatar or video in IVR so this will be explored in more detail during the optimisation phase of intervention development [11]. Similarly, although participants did not have the opportunity to embody and observe their own avatar or others', when asked about the idea of interacting with other avatars in IVR, some were sceptical that the experience would not be as meaningful as reality due to a lack of social cues, such as moving lips. This was also emphasised in previous research exploring social IVR where these social cues were missing, affecting participants' experiences with other avatars negatively [39]. However, it is now possible to integrate features such as lip-syncing into avatars [40], enhancing the social experience for IVR participants. As such, it is another example of the importance of clarifying any misconceptions or uncertainty IVR participants might have prior to use.

Participants offered mixed responses to the intervention content presented to them. For example, the use of strategies to encourage and maintain the use of IVR for reducing sedentarism, such as goal setting and monitoring, were met with enthusiasm by some but for others, they were perceived as potential hindrances to the experience–making it a chore rather than something to look forward to. This view is also present in previous research exploring motivators and barriers to reducing sedentary behaviour in older adults [41]. Qualitative feedback from participants in this study highlighted that using strategies to encourage them to reduce their time spent sedentary would feel "artificial and false". The evidence suggests that older adults need to feel that there is purpose or meaning to the strategy if they are to adopt it in their everyday lives.

Participants' beliefs about social, physiological, and psychological factors influenced how they responded both to information about sedentary behaviour as well as their experiences with IVR. Social interaction was a pattern present throughout the analysis, with participants in favour of learning how to use IVR with more experienced people and other retired and non-working adults, as well as showing interest in taking part in IVR activities with others. This concurs with evidence synthesised in a systematic review exploring adults' experiences with non-workplace sedentary behaviour interventions, where friends and family appear to act as a prompt, reminder, and motivation to take part in non-sedentary activities [42]. Participants also offered insights about their age and how their lifestyle is changing. A common pattern across the literature is older adults' adoption of new activities which are suited to their ageing bodies; which generally meant less active, more sedentary pastimes than when they were younger [42]. With regards to certain misconceptions, some participants believed that the prolonged length of time they spend sedentary each day is not problematic as they believe they are exercising regularly outside of this activity. This same misconception, or distortion, can be seen in previous research [41], with systematic review evidence suggesting that this may be due to a lack of education regarding the effects of sedentary behaviour and how it differs from physical inactivity [42]. The argument that moderate to vigorous physical activity cannot mitigate the negative health outcomes of prolonged sedentary activity alone has been contested to a degree in recent years, however, with a recent harmonised meta-analysis including more than 44,000 middle-aged and older adults reporting that "about 30–40 min of MVPA per day attenuate the association between sedentary time and risk of death. . ." [43]. With this in mind, the information provided to people about sedentary behaviour must be considered further in light of this emerging evidence–emphasising the added importance of physical activity and the positive health outcomes associated with it.

## Implications and recommendations

The aim of this study was to explore older adults' perspectives on the STAND-VR intervention content prior it's integration into a virtual environment. As presented in the supporting information section (see S2 Table and S3 Table), based on participant feedback, practical improvements will be made to the STAND-VR intervention content and integrated into a virtual environment.

With regards to presenting health information to retired and non-working adults, a tailored approach appears to be necessary. Participants provided a variety of health information delivery preferences (see S4 Table), with some preferring video and audio to text and vice versa, for example. It is important to offer access to preferred health information delivery formats as participant responses suggest they are more likely to engage with some information platforms over others. In addition, and in line with previous findings [42], it is essential to convey health information about sedentary behaviour in a clear and accessible way and from a reliable source to reduce the likelihood of any confusion between time spent sedentary and time spent physically active. This will be accounted for in the next iteration of the STAND-VR intervention, with these various formats of health information delivery being explored during the next data collection phase.

In the context of existing research, there were several common patterns identified across the data analysed in this study and the findings from other studies. There appears to be a need to focus on what is meaningful to retired and non-working adults, rather than pushing the importance of the health benefits of certain activities [35,42]–which is in line with the chosen target behaviour of this intervention. This is important when designing health behaviour change interventions in the future as it suggests the framing of the intervention should be to emphasise what meaning the end-user can derive, beyond what health benefits it offers. Although physical health outcomes, specifically long-term health outcomes associated with reduced sedentary activity, are also important to this cohort, other experiences such as enhanced social connection and enjoyment need to be part of the activities that lead to these health outcomes.

The findings also point to the importance of empowering people through health behaviour change interventions. This can be seen through the mixed responses to strategies to reduce their time spent sedentary. Although some participants were open to the idea of using strategies such as goal setting and monitoring of their sedentary behaviour, others did not like the intrusive nature of these strategies in their lives. Participants wanted the choice to take or leave such strategies rather than it being prescribed to them. As such, we recommend offering agency in behaviour change interventions, where participants have a say in how such strategies, in this case in the form of behaviour change techniques, are applied and how they are used by them–which, similar to the above point, can empower end-users to derive more personal meaning from the intervention [41]. This agency further enables users to scrutinise how digital health interventions are personalised for them [44], bringing the role of the technology in their lives to the front of their minds, empowering them to reflect more critically on how and why they are using a technology such as IVR. This will be explored further in future work related to this intervention.

The value of social support when learning a new skill and changing a health behaviour was present in many of the interviews conducted and has important implications for this research. There was a substantial preference to have someone present to help participants set up the IVR equipment, although they found using the technology quite easy, regardless. The reassurance that they were safe provided by the interviewer likely influenced their ability to focus on learning how to use the technology–a preference which many participants shared in the post-

interview sessions. Some participants suggested further that they would be interested in learning how to use IVR with other retired and non-working adults as they saw that as a supportive learning environment. Although IVR offers people the opportunity to take part in activities together from different locations, the initial learning experience seems to be one that retired and non-working adults would enjoy together.

## Strengths and limitations

This qualitative interview study was conducted following Braun and Clarkes' six steps to reflexive thematic analysis [21], paying close attention to recent updates published by the authors on how to apply this approach in qualitative research [27,45,46]. The lead author engaged in reflexivity throughout the research process to offer further justification for decisions made regarding how data was collected, analysed, and reported. Supplementary files such as reflexive journal entries are pre-registered on the lead author's open science framework repository.

Participants did not have the opportunity to design or observe full-body avatars during their IVR experience, making their responses to questions relating to avatar use purely hypothetical. This process will be explored further during the next data collection phase, where participants will have the opportunity to choose their own pre-designed avatar to embody as well as the opportunity to interact in a virtual environment with other retired and non-working adults. The demographic of the sample of participants who took part in this study was also relatively homogenous (see Tables 3 and 4). All participants described themselves as white Irish, the majority of whom had either third level education or post-secondary school professional training. Many participants also shared that the cost of the equipment would not be an issue. Although these perspectives are valuable, further research is needed to understand how people with other socio-economic capacities experience IVR and its potential impact on their sedentary activity [47].

## Conclusions

The findings of this study offer further insight into retired and non-working adults' perspectives on IVR. Moving forward, these findings can contribute to future work which aims to design IVR experiences that are more accessible to retired and non-working adults, offering greater agency to take part in activities that reduce sedentary behaviour and improve associated health outcomes and, importantly, offer further opportunity to take part in activities they can ascribe greater meaning to.

## Supporting information

**S1 Text. Interview Schedule.**
(DOCX)

**S2 Text. Demographic Questionnaire.**
(DOCX)

**S1 Table. IPAQ Data.**
(DOCX)

**S2 Table. Design Considerations.**
(DOCX)

**S3 Table. Intervention Content Suggestions.**
(DOCX)

**S4 Table. Health Information Delivery Preferences.**
(DOCX)

## Acknowledgments

I would like to thank patient and public involvement contributors, Tony Mannion and Geraldine Kenny, for the consultations they provided throughout this research study. I would like to thank Gearóid Reilly and Aisling Flynn for their collaboration in developing the VR FOUNDations training environment used in this study. For the purpose of Open Access, the author has applied a CC BY public copyright licence to any Author Accepted Manuscript version arising from this submission.

## Author Contributions

**Conceptualization:** David Healy, Emma Carr, Owen Conlan, Jane C. Walsh.

**Data curation:** David Healy.

**Formal analysis:** David Healy.

**Funding acquisition:** Owen Conlan, Jane C. Walsh.

**Investigation:** David Healy.

**Methodology:** David Healy, Emma Carr.

**Project administration:** David Healy.

**Resources:** David Healy.

**Supervision:** Owen Conlan, Jane C. Walsh.

**Writing – original draft:** David Healy.

**Writing – review & editing:** David Healy, Emma Carr, Owen Conlan, Anne C. Browne, Jane C. Walsh.

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
