## [Decision Letter · Decision Letter 0]

27 Jan 2023

PDIG-D-22-00281

Exploring the content of the STAND-VR intervention: a qualitative interview study

PLOS Digital Health

Dear Dr. Healy,

Thank you for submitting your manuscript to PLOS Digital Health. After careful consideration, we feel that it has merit but does not fully meet PLOS Digital Health's publication criteria as it currently stands. Therefore, we invite you to submit a revised version of the manuscript that addresses the points raised during the review process.

The manuscript is of high quality, so there were only a few comments raised in the review process, which are displayed below.

Please submit your revised manuscript within 30 days Feb 26 2023 11:59PM. If you will need more time than this to complete your revisions, please reply to this message or contact the journal office at digitalhealth@plos.org. Please include the following items when submitting your revised manuscript:

We look forward to receiving your revised manuscript.

Kind regards,

Laura M. König

Academic Editor

PLOS Digital Health

Journal Requirements:

1. Please send a completed 'Competing Interests' statement, including any COIs declared by your co-authors. If you have no competing interests to declare, please state "The authors have declared that no competing interests exist". Otherwise please declare all competing interests beginning with the statement "I have read the journal's policy and the authors of this manuscript have the following competing interests:"

3. We ask that a manuscript source file is provided at Revision. Please upload your manuscript file as a .doc, .docx, .rtf or .tex.

4. We have noticed that you have uploaded Supporting Information files, but you have not included a list of legends. Please add a full list of legends for your Supporting Information files after the references list. 

Additional Editor Comments (if provided):

Reviewers' comments:

Reviewer's Responses to Questions

**Comments to the Author**

1. Does this manuscript meet PLOS Digital Health’s publication criteria? Is the manuscript technically sound, and do the data support the conclusions? The manuscript must describe methodologically and ethically rigorous research with conclusions that are appropriately drawn based on the data presented.

Reviewer #1: Yes

2. Has the statistical analysis been performed appropriately and rigorously?

Reviewer #1: Yes

3. Have the authors made all data underlying the findings in their manuscript fully available (please refer to the Data Availability Statement at the start of the manuscript PDF file)?

Reviewer #1: Yes

4. Is the manuscript presented in an intelligible fashion and written in standard English?

Reviewer #1: Yes

5. Review Comments to the Author

Reviewer #1: Dear authors,

In my opinion the quality of both the manuscript and research are excellent and will add valuable information to this area of research. 

I only have a few very minor suggestions which could help to improve the manuscript.

I personally would be interested in the practical applications of your findings, especially the intervention that will or had been developed based on this research.

- Author Summary

p.3 “Prolonged sedentary behaviour, described as 6 or more hours…”

I am personally unfamiliar with this definition of prolonged sedentary behaviour (SB). Prolonged sedentary time, usually refers to time spent in SB without breaks in sedentary time, i.e. sitting for more than 30/60min etc.

- Introduction

p.3 “Prolonged sedentary behaviour has been identified as a potential independent contributor to a number of chronic conditions as well as mortality (2–4).” 

See above.

p.3-4 “epidemiological evidence suggests that 6 or more hours of sedentary behaviour per day is associated with numerous morbidities and all-cause mortality as well as significant costs to public healthcare services (6,7). Systematic review evidence indicates that many older adults are spending greater than 6 hours per day in a sedentary position (7,8), bringing them over the threshold for potential health risks that could impede their overall health and wellbeing into old age.”

For readers of this manuscript, an estimate of older adults’ objectively monitored SB would be more informative here: self-report measurements typically drastically underestimate time spent in SB as stated by the review the authors cited (8). Harvey et al. 2015 found that "Objective measurement of SB shows that older adults spend an average of 9.4 hr a day sedentary, equating to 65-80% of their waking day. Self-report of SB is lower, with average weighted self-reports being 5.3 hr daily."

p. 5 “This intervention content offers additional opportunities for retired and non-working adults to reduce the prolonged periods of time they spend sedentary each day using virtual reality.”

In order to avoid confusion concerning the use of the word “prolonged” (see above prolonged SB), the authors could provide a brief definition of those prolonged bouts.

- Discussion

p.28 “With regards to certain misconceptions, some participants believed that the prolonged length of time they spend sedentary each day is not problematic as they believe they are exercising regularly outside of this activity. This same misconception, or distortion, can be seen in previous research (39), with systematic review evidence suggesting that this may be due to a lack of education regarding the effects of sedentary behaviour and how it differs from physical inactivity (40).” 

I disagree to some degree with the authors conclusion. Findings regarding this specific topic have been contradictory. More recently Ekelund and colleagues (2020) for example have found that the deleterious effects of sedentary behaviour might be attenuated by engaging in moderate to vigorous physical activity: “About 30–40 min of MVPA per day attenuate the association between sedentary time and risk of death, which is lower than previous estimates from self-reported data.” (Ekelund, U., Tarp, J., Fagerland, M. W., Johannessen, J. S., Hansen, B. H., Jefferis, B. J., . . . Lee, I.-M. (2020). Joint associations of accelero-meter measured physical activity and sedentary time with all-cause mortality: a harmonised meta-analysis in more than 44 000 middle-aged and older individuals. British Journal of Sports Medicine, 54(24), 1499-1506. doi:10.1136/bjsports-2020-103270)

6. PLOS authors have the option to publish the peer review history of their article (what does this mean?). If published, this will include your full peer review and any attached files.

**Do you want your identity to be public for this peer review?** For information about this choice, including consent withdrawal, please see our Privacy Policy.

Reviewer #1: No

---

## [Editor Report · Decision Letter 1]

7 Feb 2023

Exploring the content of the STAND-VR intervention: a qualitative interview study

PDIG-D-22-00281R1

Dear Healy,

We are pleased to inform you that your manuscript 'Exploring the content of the STAND-VR intervention: a qualitative interview study' has been provisionally accepted for publication in PLOS Digital Health.

Best regards,

Laura M. König

Academic Editor

PLOS Digital Health